# Insulin-like Growth Factor 1, Growth Hormone, and Anti-Müllerian Hormone Receptors Are Differentially Expressed during GnRH Neuron Development

**DOI:** 10.3390/ijms241713073

**Published:** 2023-08-22

**Authors:** Alyssa J. J. Paganoni, Rossella Cannarella, Roberto Oleari, Federica Amoruso, Renata Antal, Marco Ruzza, Chiara Olivieri, Rosita A. Condorelli, Sandro La Vignera, Fationa Tolaj, Anna Cariboni, Aldo E. Calogero, Paolo Magni

**Affiliations:** 1Department of Pharmacological and Biomolecular Sciences, Università degli Studi di Milano, 20133 Milan, Italy; alyssa.paganoni@unimi.it (A.J.J.P.); roberto.oleari@unimi.it (R.O.); federica.amoruso@unimi.it (F.A.); antalrenata87@gmail.com (R.A.); ruzzamarco95@gmail.com (M.R.); chia.olivieri@gmail.com (C.O.); fationa.tolaj@unimi.it (F.T.); paolo.magni@unimi.it (P.M.); 2Department of Clinical and Experimental Medicine, University of Catania, 95123 Catania, Italy; rosita.condorelli@unict.it (R.A.C.); sandrolavignera@unict.it (S.L.V.); acaloger@unict.it (A.E.C.); 3Glickman Urological & Kidney Institute, Cleveland Clinic Foundation, Cleveland, OH 10681, USA; 4IRCCS MultiMedica, Sesto S. Giovanni, 20099 Milan, Italy

**Keywords:** GnRH, AMHR2, GHR, IGF1R, neuron migration, GnRH secretion, hypogonadotropic hypogonadism

## Abstract

Gonadotropin-releasing hormone (GnRH) neurons are key neuroendocrine cells in the brain as they control reproduction by regulating hypothalamic-pituitary-gonadal axis function. In this context, anti-Müllerian hormone (AMH), growth hormone (GH), and insulin-like growth factor 1 (IGF1) were shown to improve GnRH neuron migration and function in vitro. Whether AMH, GH, and IGF1 signaling pathways participate in the development and function of GnRH neurons in vivo is, however, currently still unknown. To assess the role of AMH, GH, and IGF1 systems in the development of GnRH neuron, we evaluated the expression of AMH receptors (AMHR2), GH (GHR), and IGF1 (IGF1R) on sections of ex vivo mice at different development stages. The expression of AMHR2, GHR, and IGF1R was assessed by immunofluorescence using established protocols and commercial antibodies. The head sections of mice were analyzed at E12.5, E14.5, and E18.5. In particular, at E12.5, we focused on the neurogenic epithelium of the vomeronasal organ (VNO), where GnRH neurons, migratory mass cells, and the pioneering vomeronasal axon give rise. At E14.5, we focused on the VNO and nasal forebrain junction (NFJ), the two regions where GnRH neurons originate and migrate to the hypothalamus, respectively. At E18.5, the median eminence, which is the hypothalamic area where GnRH is released, was analyzed. At E12.5, double staining for the neuronal marker ß-tubulin III and AMHR2, GHR, or IGF1R revealed a signal in the neurogenic niches of the olfactory and VNO during early embryo development. Furthermore, IGF1R and GHR were expressed by VNO-emerging GnRH neurons. At E14.5, a similar expression pattern was found for the neuronal marker ß-tubulin III, while the expression of IGF1R and GHR began to decline, as also observed at E18.5. Of note, hypothalamic GnRH neurons labeled for PLXND1 tested positive for AMHR2 expression. Ex vivo experiments on mouse sections revealed differential protein expression patterns for AMHR2, GHR, and IGF1R at any time point in development between neurogenic areas and hypothalamic compartments. These findings suggest a differential functional role of related systems in the development of GnRH neurons.

## 1. Introduction

The neuropeptide gonadotropin-releasing hormone (GnRH) plays a key role in the regulation of mammalian reproduction [1]. It is secreted by GnRH neurons, which originate in the olfactory placode and, in close association with the terminal nerve, migrate from the medial part of the nasal epithelium to the forebrain in the sixth embryonic week in humans [2,3]. After reaching the telencephalon, they penetrate the brain caudal to the olfactory bulb and migrate from the medial wall of the cerebral hemisphere to the preoptic/hypothalamic region, where they finally settle [3]. An alteration of this pathway leads to congenital hypogonadotropic hypogonadism (CHH) [4]. CHH is a disease caused by the abnormal production, secretion, or function of GnRH. Population-based epidemiological studies report a CHH prevalence of 1 in 48,000 persons [5]. It is more common in men, with a male-to-female prevalence of 3–5:1 [6,7]. Clinical characteristics of CHH include incomplete or absent puberty and, in adulthood, isolated hypogonadotropic hypogonadism (HH) and infertility. Many patients are diagnosed in late adolescence or early adulthood, since differential diagnosis with puberty delay is challenging during early adolescence. Over the past few decades, more than 50 genes have been found to play a role in the pathogenesis of CHH [8,9]. Furthermore, the traditional monogenic inheritance of CHH has been revised, and oligogenic forms of CHH have replaced the Mendelian view of inheritance, thus changing genetic testing and patient counseling [10]. Despite these discoveries, to date, in about half of CHH cases, the etiology remains unknown [11]. Therefore, a better understanding of the molecular bases of GnRH neuron development and function and their pathophysiological consequences is needed to help clarify the causes of idiopathic CHH.

Recently, the anti-Müllerian hormone (AMH), growth hormone (GH), and insulin-like growth factor 1 (IGF1) systems have been found to play a relevant role in the development and function of the GnRH neuronal system. Emerging evidence attributes a possible role to AMH in enhancing GnRH secretion. In particular, the intracerebroventricular injection of recombinant AMH stimulates the firing of GnRH neurons and, therefore, the secretion of GnRH in 4–8-month-old female mice [12]. In line with the possible role of AMH in CHH, the AMH receptor (AMHR2) has been found expressed in GnRH neurons of human three- and nine-week-old fetuses (at this embryonal stage, GnRH neurons are found in the nasal region at the beginning of the migratory pathway), and in the adult human hypothalamus of a man and a woman, post-mortem [12]. Furthermore, in humans, heterozygous mutations with loss of function of the *AMH* or *AMHR2* genes have been described in 3% of probands with CHH [13]. These data support that the AMH/AMHR system is involved in GnRH neuron function and, therefore, may play a role in the etiology of CHH, although, the evidence is inconclusive.

Fewer data are available on the effect of GH and IGF1 on GnRH neuron function. Delayed puberty or an increase in the duration of the entire puberty process is frequently reported in patients with Laron syndrome who have a resistance to GH due to mutations in the *GHR* gene [14,15,16,17]. This suggests that the GH-IGF1 system is involved in the timing of the activation of GnRH firing. Consistent with this hypothesis, IGF1R is expressed in GnRH neurons in both male and female mice [18], and the incubation of GT1-7 cells (an established model of mature GnRH neurons [19]) with IGF1 has been shown to increase GnRH secretion after 2 h and decrease after 4 h of incubation [20].

We have recently shown that GH, IGF1, and AMH modulate several functions of immortalized models of GnRH neurons [21]. The expression pattern of their receptors during the whole process of GnRH neuron development clearly represents the mandatory molecular background for such activities; however, it has been only partially characterized. Thus, in this paper, we comprehensively studied the expression of GHR, IGF1R, and AMHR2 proteins during mouse embryogenesis in anatomical regions relevant to GnRH neuron development with the aim of filling this knowledge gap.

## 2. Results

To investigate the potential role of AMHR2, GHR, and IGF1R during the development of GnRH neurons, we performed double immunofluorescence analyses using pan-neuronal and specific GnRH neuron markers. During mouse embryogenesis, GnRH neurons originate in the nasal placode arising from the VNO from E10.5 onwards [22]. Subsequently, they migrate into the nasal parenchyma following the vomeronasal (VN) and olfactory (OLF) axons to cross the cribriform plate (CP), enter the brain, and reach the medial preoptic area (MPOA) of the hypothalamus, where they arrest and send projections to the median eminence (ME) [23] at around E18.5.

### 2.1. IGF1R, GHR, and AMHR2 Are Expressed in Neurogenic Niches of the Olfactory and Vomeronasal Systems during Early Embryo Development

We initially analyzed the expression pattern of IGF1R, GHR, and AMHR2 in mouse embryo heads at E12.5, when, in the mouse, VN and GnRH neuron cell bodies can be visualized in the germinative epithelium of the VNO. At this stage, it is also possible to observe migrating cells forming the so-called migratory mass, which includes GnRH neurons as well as extending VN axons along the nasal septum [24]. Thus, we performed double immunofluorescence staining on coronal sections from E12.5 mouse heads, using antibodies for IGF1R, GHR or AMHR2 in combination with an antibody for nTUBB3, a pan-neuronal marker for immature neurons as well as migrating neurons/extending axons along the nasal septum. As depicted in Figure 1, IGF1R was found to be widely expressed by cell bodies located in the VNO and in the olfactory epithelium (OE), where VN and GnRH neurons and olfactory neurons are located, respectively; furthermore, IGF1R expression was also present in a bundle of neurons/axons emerging from the VNO and the OE.

GHR was expressed by sparse cells in VNO/OE neurogenic epithelia and on cells/axons emerging from the VNO, with prominent expression in cells within axon bundles at CP level, just below the forebrain (FB) (Figure 2).

Finally, AMHR2 showed a pattern of expression similar to the other receptors, with a clear colocalization within nTUBB3^+^ cell bodies in the VNO/OE and cells/axons leaving the VNO; in addition, AMHR2 expression was also found in cells at the CP level (Figure 3).

These results collectively indicate that IGF1R, GHR, and AMHR2 might be involved in the early phases of development of the olfactory/VN systems, and in particular for the targeting of nasal axons and the migration of neurons emerging from the VNO.

### 2.2. IGF1R and GHR Are Expressed by GnRH Neurons Migrating in the Nasal Parenchyma

To ascertain whether among the AMHR2, GHR, and IGF1R-positive cells in the VNO, there are GnRH^+^ neurons, we performed a double immunofluorescence protocol with an antibody raised against GnRH peptide [25,26]. Although the expression of both IGF1R and GHR in the VNO epithelium has been observed, as reported above, GnRH neurons in the VNO did not express these receptors. Yet, we found that IGF1R and GHR were expressed by some GnRH-positive cells exiting the VNO within the nasal parenchyma (Figure 4).

Due to the cross-reactivity of primary antibodies, we could not perform similar colocalization staining for AMHR2 and GnRH. In any case, previous work has already reported the expression of AMHR2 by migrating GnRH neurons both in mouse embryos at E12.5 and human fetuses at gestation week 9 [12]. As suggested by these double immunofluorescence staining, some GnRH-positive neurons, once emerged from the VNO epithelium, express IGF1R and GHR, indicating that these receptors could be implicated in the initiation of GnRH neuron migration in the nasal parenchyma.

### 2.3. IGF1R, GHR, and AMHR2 Expression Is Maintained in Nasal Axons and Migrating Cells at the Intermediate Embryo Developmental Stage

We then analyzed the expression pattern of AMHR2, GHR, and IGF1R in mouse embryo heads at E14.5, an intermediate time point, at which migrating GnRH neurons can be found in both the nose and in the FB [23]. Double immunostaining with antibodies for these receptors and nTUBB3 confirmed that AMHR2, IGF1R, and GHR were expressed on nTUBB3-positive cells/axons migrating/extending in the nasal compartment. In particular, IGF1R was still highly expressed in the dorsal portion of the OE, in the VNO, and by the migrating cells/elongating axons in the nasal parenchyma (Figure 5A,B). Furthermore, IGF1R was expressed by the nasal axons that innervate the OBs, at the level of CP (Figure 5C,D).

Similarly, GHR was found in migrating cells/nasal axons emerging from the VNO and on axons reaching the OBs at the level of the CP, but its expression was less pronounced in the neurogenic epithelia (OE and VNO) compared to IGF1R. In addition, GHR expression is also reported on nasal septum and nasal capsule cells (Figure 6).

Finally, we found that AMHR2 was expressed in the VNO and dorsal OE epithelial cells and by migrating cells/extending axons emerging from the VNO. Moreover, AMHR2 was also expressed on the axons innervating the OBs (Figure 7).

Overall, these results strongly suggest a supportive role for AMHR2, GHR, and IGF1R in the olfactory/VN axon patterning and/or migration of cells emerging from VNO, such as GnRH neurons.

### 2.4. AMHR2 Expression Is Maintained by GnRH Neurons through Their Developmental Trajectory, Whereas GHR and IGF1R Expression Follows a Decreasing Pattern

We then investigated whether these receptors were also expressed by GnRH neurons themselves at E14.5, when GnRH neuron migration is at its peak and shows cells present in all the three compartments (nasal area, cribriform plate, MPOA), and at E18.5 when GnRH neurons are settled in the hypothalamic MPOA. Also, in these experiments, we combined immunostaining for IGF1R and GHR with a specific anti-GnRH antibody. For double immunostaining with AMHR2, we employed an antibody directed against PLXND1, which allows the selective detection of GnRH neurons in the MPOA [27,28]. Since PLXND1 in the nasal area is highly expressed by OLF/VN axons, we could not use this antibody to confirm colocalization with GnRH neurons in this district; however, previous studies have already demonstrated that AMHR2 is expressed in the nasal compartment at E12.5 [12].

Due to the presence of dense IGF1R^+^/TUJ1^+^nasal axon bundles in the nasal parenchyma at this stage, it is difficult to conclusively ascertain IGF1R-GnRH colocalization in this area. Moreover, when cells reached the CP, no colocalization was observed between these two proteins (Figure 8A–D). Since starting from E14.5 GnRH neurons are also scattered throughout MPOA, we performed a similar analysis in the developing hypothalamus at E14.5 and E18.5. We found that MPOA-resident GnRH neurons did not express IGF1R at both time points analyzed (Figure 8E–H). Of note, a high IGF1R signal was detected in close proximity to the organum vasculosum of the lamina terminalis (OVLT) (Figure 8H).

In addition, consistent with the findings obtained from earlier developmental stages, GHR was still expressed by some GnRH neurons emerging from the VNO (Figure 9A,B), but not by GnRH neurons reaching the CP, nor neurons projecting to the MPOA (Figure 9C–F). As a confirmation of a role of this receptor during the early phases of GnRH neuron migration and nasal axon patterning, GHR expression was absent on scattered GnRH neurons at E18.5 in the MPOA (Figure 9G,H).

Finally, we performed similar double immunofluorescence experiments for AMHR2 and PLXND1 at E14.5 and found colocalization signals in some PLXND1^+^ neurons at this stage (Figure 10A,B). Furthermore, at E18.5, when GnRH neurons project to the MPOA and to the OVLT and ME, the AMHR2 signal was still detected in PLXND1^+^ neurons (Figure 10C,D).

## 3. Discussion

CHH is a pathological condition characterized by a strongly represented genetic background, with more than 30 genes found mutated in patients, so far. The underlying genetics is complex and it is now recognized that more than one gene can be found mutated in the same patient. Several biological mechanisms have also been proposed as the etiology of this disorder, including the impaired migration of GnRH neurons into the hypothalamus, abnormal GnRH secretion, and lack of GnRH biofunctional activity [8]. Despite advances in the field of genetic diagnostics, which by exploiting genomic techniques allow the identification of a large panel of mutations through next-generation sequencing, the etiology of CHH is still not understood in a rather significant number of affected patients. A recent single-center cohort study of 388 CHH patients with no evidence of any acquired causes of HH reported a genetic etiology in just ~36% of patients, who were studied using a panel of 28 genes. Therefore, the exact mechanism(s) leading to the onset of the disease remain unknown in most patients [29]. This requires further studies to understand whether any other pathogenetic mechanisms are involved in the (dis)regulation of the function of GnRH-secreting neurons.

In the present study, the expression of AMHR2, GHR, and IGF1R was explored in the GnRH neuronal system and in the olfactory/VN systems during key time points in the development of GnRH-secreting neurons in mice. The structures chosen for the study interact functionally to ensure the correct migration and patterning of GnRH neurons in the mouse. In particular, the expression of these receptors was studied in the neurogenic niches of the VNO and OE at E12.5, in the nasal parenchyma at E14.5, when the pattern of nasal axons and the migration of GnRH neurons are at their peak, and in MPOA between E14.5 and E18.5, when GnRH neurons settle down before projecting their nerve terminal to the ME. The analysis of the expression of these receptors was combined with that of specific markers for GnRH neurons, as well as for other cellular components such as the neurogenic niches of the VNO and olfactory epithelium, and the patterning of emerging VN nerves and olfactory axons.

In our previous study, we demonstrated that these three receptors are expressed in vitro by immortalized immature and mature models of mouse GnRH neurons [21]. Here, we found that AMHR2, IGF1R, and GHR are expressed, during embryo-fetal life in the mouse, by early-migrating GnRH neurons. AMHR2 is expressed in the final phases of GnRH migration, whereas the expression of IGF1R and GHR decreases with progression of developmental stages and differentiation of GnRH neuron. These findings hint at a role for these receptors in the initial phases of migration and suggest a role for AMHR2 in the final phases of GnRH neuron settlement in the hypothalamus, in which, instead, IGF1R and GHR appear to play a superfluous role.

Recent evidence in experimental animal studies intriguingly suggests a role of AMH and its receptor in the function of GnRH neurons. Indeed, Cimino and colleagues induced the activation of these neuronal cell firings after the intracerebroventricular injection of AMH, indicating the ability of this hormone to stimulate GnRH release [12]. In the same study, the authors reported post-mortem AMHR2 expression in the adult hypothalamus of a man and a woman, as well as three- and nine-year-old human fetuses [12]. Although this evidence may suggest a modulatory role of AMH in the secretion of GnRH neurons, scanty data are available on the ontogenic effects of these genes on GnRH-secreting neurons. Therefore, the present study extends and strengthens the knowledge of the role of AMH/AMHR2 on the migration and function of GnRH neurons, making the possible clinical implications of the AMH/AMHR2 pathway increasingly concrete, especially in explaining the etiology of some CHH forms. Consistent with this, Malone and colleagues pioneered the presence of heterozygous loss-of-function mutations of the *AMH* and *AMHR2* genes in 3% of CHH probands using whole exome sequencing [13].

As for the complex relationship between the GH/IGF1 axis and GnRH-secreting neurons, this appears to be an even less well-known topic. Some authors have hypothesized a role of IGF1 in the physiology of the onset of puberty, due to the “acromegalic” levels of this hormone during the pubertal phase [30]. In support of this concept, delayed puberty was recently described in a patient with IGF1R defects [31]. Furthermore, this is a common phenotype in patients with Laron syndrome, a genetic disorder caused by a mutation of the *GHR* gene [32]. Further confirming the clinical findings on the relationship between GnRH neuron function and the GH/IGF1 axis, a study of 138 Chinese patients with CHH reported the presence of *GH1*, *GHR*, and *IGF1* gene variants after exome sequencing analysis [33]. Mouse GnRH-secreting neurons express the *Igf1r* gene [18] and IGF1 has previously been shown to stimulate GnRH secretion in vitro [20]. A recent study evaluated changes in GnRH neuron gene expression in mice during proestrus and metestrus, two phases that precede the surge of GnRH release triggered by the rise of 17ß-estradiol [34]. The authors reported modulation in the expression of several genes, including *Amhr2* and *Igf1r* at proestrus. Preclamp recordings during metestrus confirmed the functional role of *Igf1r* gene modulation, such as activation of this receptor-induced increase in the frequency of miniatured post-synaptic currents, leading to GnRH secretion [34]. The authors concluded by suggesting a role for these growth factors in the release of the GnRH surge [34].

Despite this evidence, to the best of our knowledge, no ex vivo study has so far evaluated the expression of GHR and IGF1R in the brains of mice along the pathway of GnRH neurons during prenatal life.

Another study reported GHR expression in rat fetuses at E12, mainly in the vascular elements (endothelium and hematopoietic cells) [35]. In the human fetus, GH can be found in fetal blood circulation as early as the 10th week of gestation, and peaks at 20–24 weeks [36]. It should be noted that the GH gene family includes five different genes: *GH1* (*GH-N*), *GH2* (*GH-V*), and three *chorionic somatomammotropin* (*CS*) genes (also known as placental lactogens), *CSH1* (*CS-A*), *CSH2* (*CS-B*), and *CSHL1*. The pituitary releases GH1 and is responsible for postnatal growth. The placenta and, in particular, the syncytiotrophoblastic layer and extravillous trophoblast cells release GH2. GH2 is the predominant form of GH in maternal circulation during pregnancy and is associated with fetal growth [37]. Thus, GHRs at E12 could be triggered by GH2.

Our IGF1R results perfectly complement previous functional studies in zebrafish, which showed that *Igf1r* blockade affects the migration and spatial organization of Gnrh3 neurons in early embryos [38]. In line with this, the *Igf1r* KO conditional GnRH mice showed a different dendritic shape of GnRH neurons (smooth vs. spiny) which, however, normalizes later. Importantly, these mice had a normal number of GnRH neurons but delayed pubertal onset, implying a relationship between impaired spatial organization and function of GnRH-secreting neurons [39]. IGF1R has been confirmed to be expressed in VNO/OE and nasal axons at E14.5, and is required for the normal projection patterns of olfactory sensory neurons [40]. Furthermore, at E18.5, we found a high expression of IGF1R around the OVLT. This is in agreement with IGF1R seen on cell bodies of GnRH neurons but only near axons in OVLT/ME, with a decreasing trend from P5 to P60 in both male and female mice [18]. Regarding the GHR, this is the first study that analyzes its expression in the developing olfactory/VN and GnRH systems ex vivo. Our results suggest a possible role of this receptor in the early migration of GnRH neurons. It will be interesting to understand the functional implications of these findings, as no studies indicate a role for GH during the development of GnRH neurons, nor in their function, in adulthood. Consequently, GHR is known to be localized in some neurons of the hypothalamic arcuate nucleus, where it is co-expressed with somatostatin, growth hormone-releasing hormone, and agouti-related peptide [41]. Its localization in MPOA/ME was, until now, unknown.

Together with already-published data from our group [21], these results may indicate a role for GH and IGF1 in facilitating the migration of GnRH neurons. Indeed, as observed for AMHR2, their receptors appear to be expressed along the pathway used by GnRH-secreting neurons during ontogeny. This could represent one of the mechanisms by which IGF1R or GHR defects are associated with delayed puberty or hypogonadism [31,33].

## 4. Materials and Methods

### 4.1. Mouse Embryos Preparation

Wild-type C57Bl6/J embryos (E12.5, E14.5, and E18.5) were purchased from Charles River (MA, USA) and used for expression studies (Italian Ministry of Health, license N° 5247B.N.QPE). To obtain embryos of defined gestational stages, mice were mated in the evening, and the morning of vaginal plug formation was counted as E0.5. Timed-pregnant females were sacrificed by cervical dislocation to harvest embryos at the desired embryonic age. The embryo heads were rapidly washed in ice-cold PBS, fixed in 4% PFA for 3 h at 4 °C and cryopreserved in 30% sucrose overnight at 4 °C. Whole embryo heads were then embedded in OCT (VWR, Radnor, PA, USA) and stored at −80 °C until further processing.

### 4.2. Double Immunofluorescence Studies

PFA-fixed 20 µm thick cryosections were processed as previously described [42]. The following primary antibodies were used: rabbit anti-AMHR2 (1:100; Invitrogen, Monza Brianza, Italy, cod. PA5-112901), goat anti-GHR (1:20; R&D Systems, Udine, Italy, cat. AF1360), goat anti-IGF1R (1:20; R&D Systems, cod. AF305-NA), rabbit anti-GnRH (1:400; Immunostar, Hudson, WI, USA, cat. 20075), goat anti-PLXND1 (1:200; R&D Systems, UD, IT, cat. AF4160), and mouse anti-Tubulin β 3 (TUBB3) (clone TUJ1, 1:200; Covance, CA, USA cat. MMS-435P). The secondary antibodies used were 488- or Cy3-conjugated donkey anti-rabbit, donkey anti-mouse, or donkey anti-goat Fab fragments (1:200; Jackson Immunoresearch, West Grove, PA, USA). Nuclei were counterstained with DAPI (1:2000; Sigma Aldrich, Saint Louis, MO, USA) before mounting with Mowiol (Sigma Aldrich).

### 4.3. Image Acquisition

Tissues were examined with a Zeiss LSM 900 confocal laser scanning microscope equipped with a Zeiss Axiocam 305 color (Zeiss, Milan, Italy) and using the following objectives: Plan-Apochromat 10× (0.45 M27), EC Plan-Neofluar 20X M-27 (NA 0.5) and Plan Apochromat 40X 1.3 Pil DIC (VIS-IR M27) objectives. DAPI, Alexa488, and Cy3 were excited at 353, 493, and 548 nm and observed at 400–500, 500–540, and 540–700 nm, respectively. 1024 × 1024 pixel images were acquired in a stepwise fashion over a defined z-focus range corresponding to all visible fluorescence within the cell. Maximum projections of the z-stack with an optical section of 0.50 μm were obtained post-acquisition using the ZEN 3.0 Suite (Zeiss). Adobe Photoshop CS6 software was used to prepare the images presented.

## 5. Conclusions

Despite advances in genetic diagnostics, the etiology of CHH remains unidentified in many cases. Therefore, it is necessary to further unveil novel pathogenetic mechanisms able to influence GnRH-secreting neurons and elucidate the etiology of idiopathic CHH. Recently, mounting evidence has suggested a role for AMH in the function of GnRH neurons. Our recent study [21] demonstrated the ability of AMH, GH, and IGF1 to enhance GN11 neuron migration and GnRH secretion from GT1-7 cell lines in vitro. To better understand the role of these hormones in the ontogeny of GnRH neurons, we analyzed AMHR2, GHR, and IGF1R expression ex vivo by immunostaining in the mouse brain at different developmental stages. We found that AMHR2, IGF1R, and GHR are expressed by the first migrating GnRH neurons, suggesting a role of these three receptors in the initial phases of migration. AMHR2 was also found expressed in the final stages of GnRH migration, whereas the expression of IGF1R and GHR decreased with time to development and the differentiation of GnRH neurons, suggesting that while AMHR2 may be required in the final phases of settlement of GnRH neurons in the hypothalamus, IGF1R and GHR may play a more marginal role in this step.

Taken together, these data could indicate that some mechanisms involved in the regulation of GnRH-secreting neuron migration may be based on the existence of a functional expression pattern of AMHR2, GHR, and IGF1R along the pathway followed by GnRH-secreting neurons during the embryonal period. To the best of our knowledge, this is the first study investigating this hitherto-unclear topic. Further research is needed to unravel the complex series of events regulating the migration of GnRH neurons, the understanding of which may be pivotal to elucidating the etiology of idiopathic forms of CHH.

## Figures and Tables

**Figure 1 ijms-24-13073-f001:**
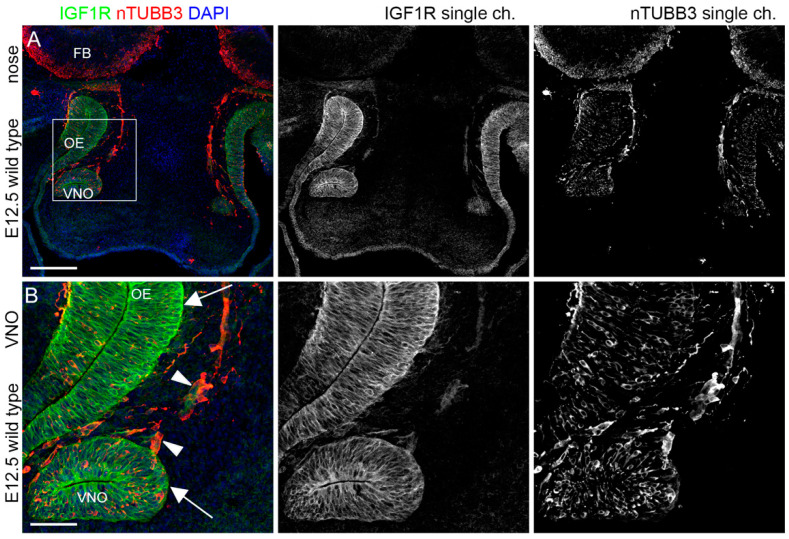
IGF1R is expressed in olfactory neurogenic niches at E12.5. (**A**) Coronal sections of E12.5 mouse embryo heads at VNO level were immunolabeled for IGF1R (green) and nTUBB3 (red) to reveal neuron cell bodies and axons. (**B**) High magnification of squared box area in (**A**). Solid arrowheads indicate examples of nTUBB3-positive cells/axons exiting the VNO that express IGF1R in the nasal parenchyma. White arrows indicate strong expression of IGF1R in neurogenic epithelia of the VNO and the OE. Single-channel images are shown beside each panel. Abbreviations: IFG1R, insulin-like growth factor 1 receptor; VNO, vomeronasal organ; OE, olfactory epithelium. Scale bars: 250 μm (**A**) and 50 μm (**B**).

**Figure 2 ijms-24-13073-f002:**
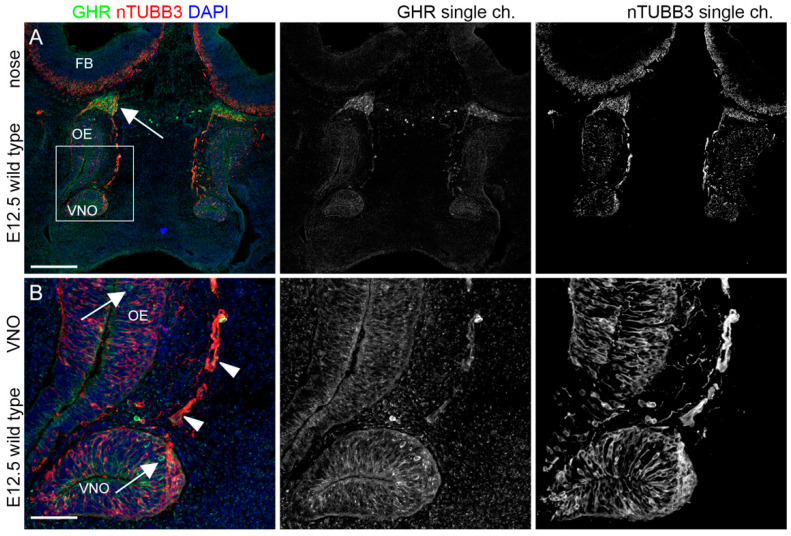
**GHR is expressed by migratory mass at E12.5.** (**A**) Coronal sections of E12.5 mouse embryo heads at the VNO level were immunolabeled for GHR (green) and nTUBB3 (red) to reveal neuron cell bodies and axons. White arrows indicate the expression of GHR in the axon bundles at the CP level. (**B**) High magnification of squared box area in (**A**). Solid arrowheads indicate examples of nTUBB3-positive cells/axons exiting the VNO that express GHR in the nasal parenchyma. White arrows indicate the expression of GHR on some cell bodies in neurogenic epithelia of the VNO and the OE. Single-channel images are shown beside each panel. Abbreviations: GHR, growth hormone receptor; VNO, vomeronasal organ; CP, cribriform plate; OE, olfactory epithelium; and FB, forebrain. Scale bars: 250 μm (**A**) and 50 μm (**B**).

**Figure 3 ijms-24-13073-f003:**
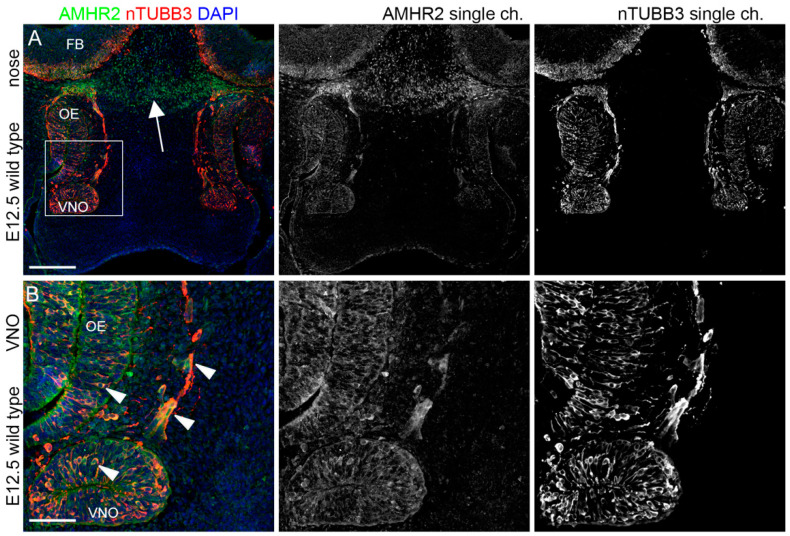
**AMHR2 is expressed in neurogenic niches of VNO, OE, and migratory mass at E12.5.** (**A**) Coronal sections of E12.5 mouse embryo heads at the VNO level were immunolabeled for AMHR2 (green) and nTUBB3 (red) to reveal neuron cell bodies and axons. White arrows indicate the expression of AMHR2 in the parenchyma at the CP level. (**B**) High magnification of squared box area in (**A**). Solid arrowheads indicate examples of nTUBB3-positive cells/axons exiting the VNO and cell bodies in neurogenic epithelia of the VNO/OE that express AMHR2. Single-channel images are shown beside each panel. Abbreviations: AMHR2, anti-Müllerian hormone receptor 2; VNO, vomeronasal organ; OE, olfactory epithelium; and CP, cribriform plate. Scale bars: 250 μm (**A**) and 50 μm (**B**).

**Figure 4 ijms-24-13073-f004:**
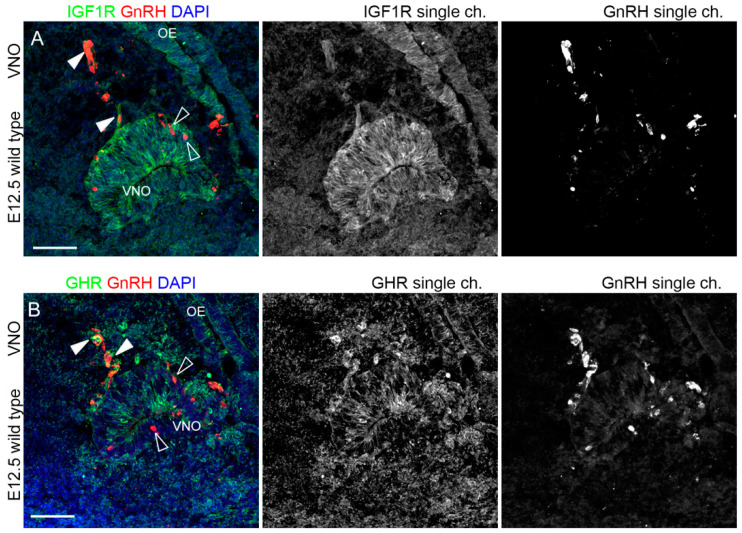
IGF1R and GHR are expressed by GnRH neurons migrating in the nasal parenchyma, but not by GnRH neurons residing in the VNO. (**A**) Coronal sections of E12.5 mouse embryo heads at the VNO level were immunolabeled for IGF1R (green) and GnRH (red) to reveal GnRH neurons. Solid arrowheads indicate examples of GnRH-positive cells exiting the VNO that express IGF1R. Empty arrowheads indicate examples of GnRH-positive cells within the VNO that do not express IGF1R. (**B**) Coronal sections of E12.5 mouse embryo heads at the VNO level were immunolabeled for GHR (green) and GnRH (red) to reveal GnRH neurons. Solid arrowheads indicate examples of GnRH-positive cells exiting the VNO that express GHR. Empty arrowheads indicate examples of GnRH-positive cells within the VNO that do not express GHR. Single-channel images are shown beside each panel. Abbreviations: IFG1R, insulin-like growth factor 1 receptor; GHR, growth hormone receptor; GnRH, gonadotropin-releasing hormone; VNO, vomeronasal organ. Scale bars: 50 μm.

**Figure 5 ijms-24-13073-f005:**
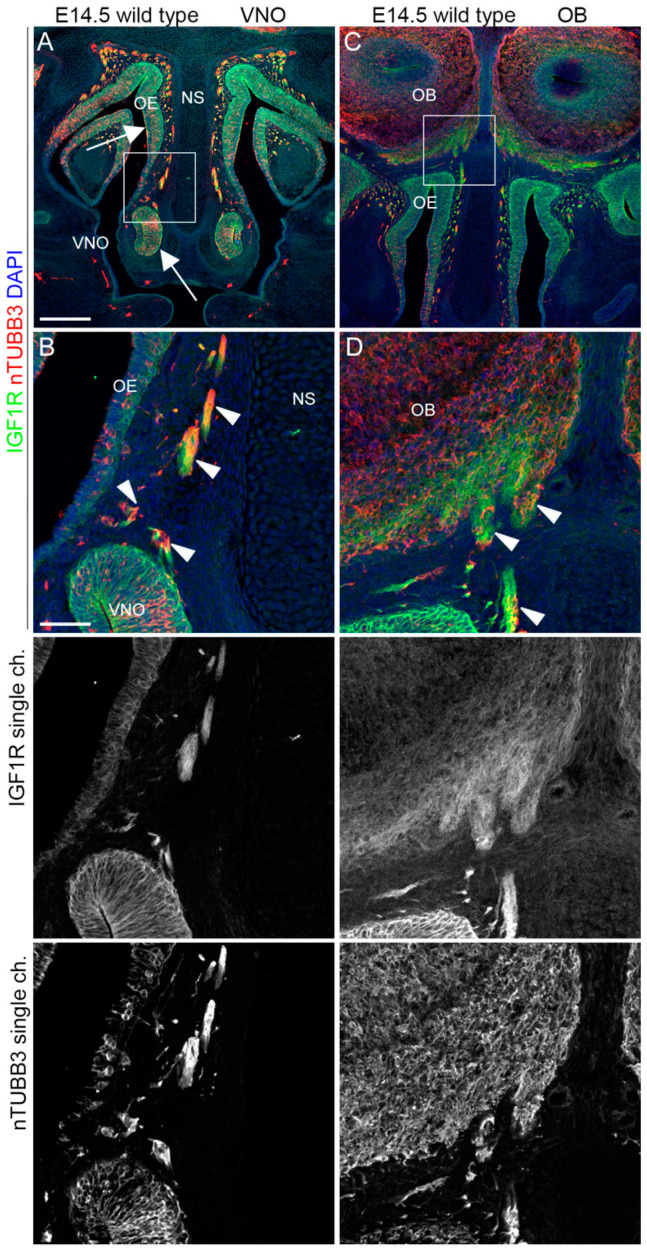
**IGF1R is expressed by neurons and axons migrating/extending from the VNO.** (**A**,**B**) Coronal sections of E14.5 mouse embryo heads at the VNO level were immunolabeled for IGF1R (green) and nTUBB3 (red) to reveal neuron cell bodies and axons. White arrows indicate the expression of IGF1R in the VNO/dorsal OE neurogenic epithelia. The high magnification of the squared box area is shown in (**B**). Solid arrowheads indicate examples of nTUBB3/IGF1R-positive cells/axons exiting the VNO. (**C**,**D**) Coronal sections of E14.5 mouse embryo heads at the OB level were immunolabeled for IGF1R (green) and nTUBB3 (red) to reveal neuron cell bodies and axons. The high magnification of the squared box area is shown in (**D**). Solid arrowheads indicate examples of nTUBB3/IGF1R-positive axons innervating the OB. Single-channel images are shown below each panel. Abbreviations: IFG1R, insulin-like growth factor 1 receptor; VNO, vomeronasal organ; OE, olfactory epithelium; OB, olfactory bulb; and NS, nasal septum. Scale bars: 250 μm (**A**,**C**) and 50 μm (**B**,**D**).

**Figure 6 ijms-24-13073-f006:**
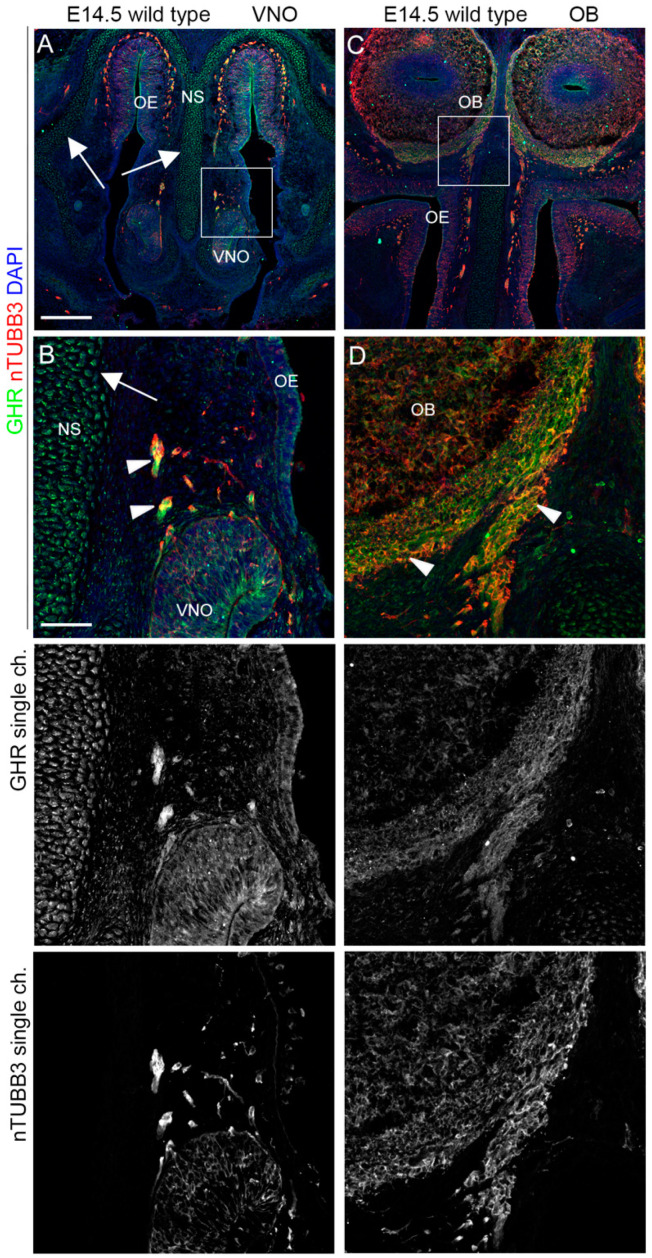
GHR is expressed by neurons and axons leaving/extending from the VNO. (**A**,**B**) Coronal sections of E14.5 mouse embryo heads at the VNO level were immunolabeled for GHR (green) and nTUBB3 (red) to reveal neuron cell bodies and axons. White arrows indicate the expression of GHR in the nasal septum and nasal capsule cartilage primordia. The high magnification of the squared box area is shown in (**B**). Solid arrowheads indicate examples of nTUBB3/GHR-positive cells/axons exiting the VNO. White arrows indicate the expression of GHR in the nasal septum. (**C**,**D**) Coronal sections of E14.5 mouse embryo heads at the OB level were immunolabeled for GHR (green) and nTUBB3 (red) to reveal neuron cell bodies and axons. The high magnification of the squared box area is shown in (**D**). Solid arrowheads indicate examples of nTUBB3/GHR-positive axons innervating the OB. Single-channel images are shown below each panel. Abbreviations: GHR, growth hormone receptor; VNO, vomeronasal organ; OE, olfactory epithelium; NS, nasal septum; OB, olfactory bulb. Scale bars: 250 μm (**A**,**C**), 50 μm (**B**,**D**).

**Figure 7 ijms-24-13073-f007:**
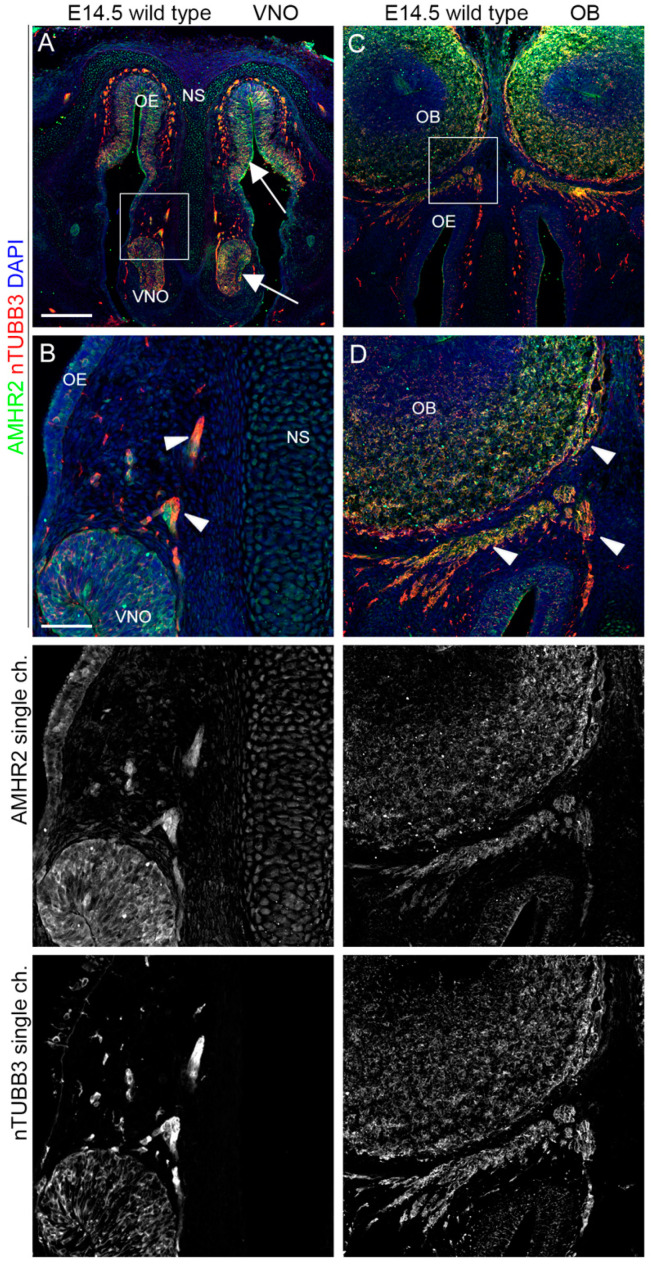
**AMHR2 is expressed by neurons and axons of the vomeronasal/olfactory systems.** (**A**,**B**) Coronal sections of E14.5 mouse embryo heads at the VNO level were immunolabeled for AMHR2 (green) and nTUBB3 (red) to reveal neuron cell bodies and axons. White arrows indicate the expression of AMHR2 in the VNO/dorsal OE neurogenic epithelia. The high magnification of the squared box area is shown in (**B**). Solid arrowheads indicate examples of nTUBB3/AMHR2-positive cells/axons exiting the VNO. (**C**,**D**) Coronal sections of E14.5 mouse embryo heads at the OB level were immunolabeled for AMHR2 (green) and nTUBB3 (red) to reveal neuron cell bodies and axons. The high magnification of the squared box area is shown in (**D**). Solid arrowheads indicate examples of nTUBB3/AMHR2-positive axons innervating the OB. Single-channel images are shown below each panel. Abbreviations: AMHR2, anti-Müllerian hormone receptor 2; VNO, vomeronasal organ; OE, olfactory epithelium; NS, nasal septum; and OB, olfactory bulb. Scale bars: 250 μm (**A**,**C**), 50 μm (**B**,**D**).

**Figure 8 ijms-24-13073-f008:**
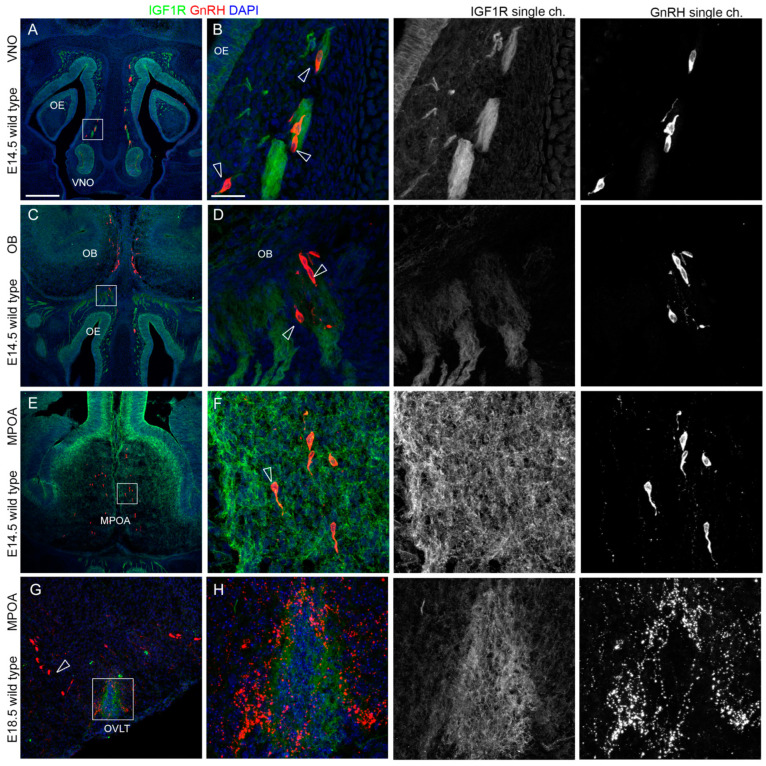
**IGF1R is not expressed by migrating and hypothalamic GnRH neurons.** (**A**,**B**) Coronal sections of E14.5 mouse embryo heads at the VNO level were immunolabeled for IGF1R (green) and GnRH (red) to reveal GnRH neurons. The high magnification of the squared box area is shown in (**B**). Empty arrowheads indicate examples of GnRH-positive cells that are in close contact with IGF1R-positive axons but do not express IGF1R themselves. (**C**,**D**) Coronal sections of E14.5 mouse embryo heads at the CP level were immunolabeled for IGF1R (green) and GnRH (red) to reveal GnRH neurons. The high magnification of the squared box area is shown in (**D**). Empty arrowheads indicate examples of GnRH-positive cells that are in close contact with IGF1R-positive axons but do not express IGF1R themselves. (**E**–**H**) Coronal sections of E14.5 (**E**,**F**) and E18.5 (**G**,**H**) mouse embryo heads at the MPOA level were immunolabeled for IGF1R (green) and GnRH (red) to reveal GnRH neurons. The high magnification of the squared box area is shown in (**F**). Empty arrowheads indicate examples of GnRH-positive cells that do not express IGF1R. Note the high expression of IGFR in the OVLT area. Single-channel images are shown beside each panel. Abbreviations: VNO, vomeronasal organ; OE, olfactory epithelium; OB, olfactory bulb; and MPOA, medial preoptic area. Scale bars: 250 μm (**A**,**C**,**E**,**G**) and 50 μm (**B**,**D**,**F**,**H**).

**Figure 9 ijms-24-13073-f009:**
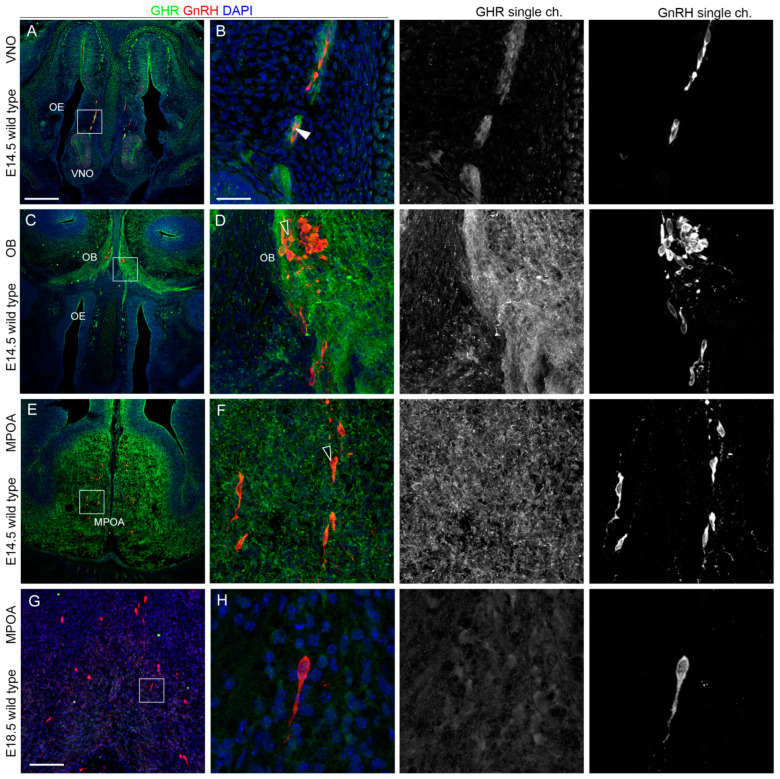
**GHR is expressed by migrating but not by hypothalamic GnRH neurons.** (**A**,**B**) Coronal sections of E14.5 mouse embryo heads at the VNO level were immunolabeled for GHR (green) and GnRH (red) to reveal GnRH neurons. The high magnification of the squared box area is shown in (**B**). Solid arrowheads indicate examples of GnRH-positive cells that express GHR. (**C**,**D**) Coronal sections of E14.5 mouse embryo heads at the OB level were immunolabeled for GHR (green) and GnRH (red) to reveal GnRH neurons. The high magnification of the squared box area is shown in (**D**). Empty arrowheads indicate examples of GnRH-positive cells that do not express GHR. (**E**–**H**) Coronal sections of E14.5 (**E**,**F**) and E18.5 (**G**,**H**) mouse embryo heads at the MPOA level were immunolabeled for GHR (green) and GnRH (red) to reveal GnRH neurons. The high magnification of the squared box area is shown in (**F**,**H**). Empty arrowheads indicate examples of GnRH-positive cells that do not express GHR. Single-channel images are shown beside each panel. Abbreviations: VNO, vomeronasal organ; OE, olfactory epithelium; OB, olfactory bulb; and MPOA, medial preoptic area. Scale bar: 250 μm (**A**,**C**,**E**,**G**,**H**), 125 μm, and 50 μm (**B**,**D**,**F**).

**Figure 10 ijms-24-13073-f010:**
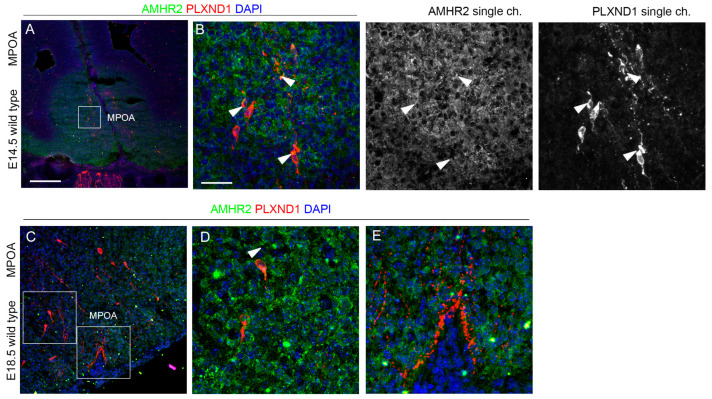
**AMHR2 is expressed by hypothalamic GnRH neurons expressing PLXND1.** (**A**–**E**) Coronal sections of E14.5 (**A**,**B**) and E18.5 (**C**–**E**) mouse embryo heads at the MPOA level were immunolabeled for AMHR2 (green) and PLXND1 (red) to reveal GnRH neurons. The high magnifications of the squared box areas are shown in (**B**–**D**). Solid arrowheads indicate examples of GnRH-positive cells that express AMHR2. Single-channel images are shown beside each panel. Abbreviations: VNO, vomeronasal organ; OE, olfactory epithelium; OB, olfactory bulb; MPOA, medial preoptic area. Scale bar: 250 μm (**A**,**C**), 50 μm (**B**,**D**,**E**).

## Data Availability

The data presented in this study are available on request from the corresponding authors.

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
