# Peer review of "Insulin-like Growth Factor 1, Growth Hormone, and Anti-Müllerian Hormone Receptors Are Differentially Expressed during GnRH Neuron Development"

_ijms, 2023, doi:10.3390/ijms241713073_

Round 1

Reviewer 1 Report

This manuscript describes a comprehensive study of the receptor population in migrating GnRH neurons during embryonic and fetal development.  Some surprising results relate to the fact that GnRH neuronal migration may be dependent on IGF-1, GH and/or AMH.  The study is well written and beautifully illustrated.  Below are two concerns. 

1)  The 4th line from bottom of abstract has a sentence that appears to have a word missing.

2) The most unusual finding was the GHR as early as E12.5.  The question that immediately comes to mind is the fact that the appearance of this receptor is several days earlier than GH from the pituitary.  That should be noted.  Also, the GH driving this receptor may not be Gh1.  it could be one of the other forms of GH.  

Author Response

Reviewer 1

This manuscript describes a comprehensive study of the receptor population in migrating GnRH neurons during embryonic and fetal development.  Some surprising results relate to the fact that GnRH neuronal migration may be dependent on IGF-1, GH and/or AMH.  The study is well written and beautifully illustrated.  Below are two concerns. 

Answer. Thank you for your words of appreciation for our study.

Comment 1. The 4th line from bottom of abstract has a sentence that appears to have a word missing.

Answer to comment 1. The sentence of the 4th line to last of the abstract has been rephrased. Changes are shown in red font. We hope it reads better now. 

Comment 2. The most unusual finding was the GHR as early as E12.5. The question that immediately comes to mind is the fact that the appearance of this receptor is several days earlier than GH from the pituitary. That should be noted.  Also, the GH driving this receptor may not be Gh1.  it could be one of the other forms of GH.  

Answer to comment 2. Thank you for this interesting suggestion. Indeed, another study reported GHR expression in rat fetuses at E12, mainly in the vascular elements (endothelium and hematopoietic cells) but the authors did not focus on the hypothalamic-pituitary axis in their study (doi: 10.1242/dev.114.4.869. PMID: 1618149). In the human fetus, GH can be found in the fetal blood circulation as early as the 10th week gestation and peaks at 20-24 weeks (doi: 10.1677/joe.0.1230003. PMID: 2681502). It should be recalled that the GH gene family includes five different genes: GH1 (GH-N), GH2 (GH-V), and three chorionic somatomammotropin (CS) genes (also known as placental lactogens): CSH1 (CS-A), CSH2 (CS-B), and CSHL1. The pituitary releases GH1 and is responsible for postnatal growth. The placenta and, in particular, the syncytiotrophoblastic layer and extravillous trophoblast cells release GH2. GH2 is the predominant form of GH in maternal circulation during pregnancy and is associated with fetal growth (doi: 10.1210/en.2018-00037. PMID: 29659791). Thus, the GHRs at E12 could be triggered by GH2.

This paragraph has now been added in the Discussion.

Reviewer 2 Report

This study examined the contribution of AMHR2, GHR, and IGF1R signaling in the development of GnRH neurons by using ex vivo double immune-fluorescent staining to detect the expression or co-expression of these receptors during mouse embryogenesis, in anatomical areas relevant to GnRH neuron development. The three receptors were localized from the early embryonic day E12.5 in the neurogenic niches of the olfactory and vomeronasal organ (VNO), with IGF1R and GHR to be expressed by VNO-emerging GnRH neurons. The expression pattern of these receptors varied in developmental time points between neurogenic areas and hypothalamic compartments, suggesting a differential functional role of these signaling  systems in the development of GnRH neurons.

The methodological approach is appropriate and enables the reader to visualize the contribution of these receptors in situ, in relevant anatomic areas of the developing mouse brain.

The results are comprehensively presented, and the Discussion is very well elaborated, so as to include previous knowledge, explain the new data and connect the experimental evidence with potential clinical relevance as in the case of congenital hypogonadotropic hypogonadism.

The question has previously been posed, but  the approach used in this manuscript is newly applied and can add to the existing knowledge.

The study is properly designed and performed with the highest technical standards; The data support the authors conclusions.

The conclusions are of interest. Apart from the basic knowledge they expand, they provide potential projections of clinical importance.

The authors address an important question with the most appropriate approach.

Author Response

Reviewer 2

This study examined the contribution of AMHR2, GHR, and IGF1R signaling in the development of GnRH neurons by using ex vivo double immune-fluorescent staining to detect the expression or co-expression of these receptors during mouse embryogenesis, in anatomical areas relevant to GnRH neuron development. The three receptors were localized from the early embryonic day E12.5 in the neurogenic niches of the olfactory and vomeronasal organ (VNO), with IGF1R and GHR to be expressed by VNO-emerging GnRH neurons. The expression pattern of these receptors varied in developmental time points between neurogenic areas and hypothalamic compartments, suggesting a differential functional role of these signaling systems in the development of GnRH neurons.

The methodological approach is appropriate and enables the reader to visualize the contribution of these receptors in situ, in relevant anatomic areas of the developing mouse brain.

The results are comprehensively presented, and the Discussion is very well elaborated, so as to include previous knowledge, explain the new data and connect the experimental evidence with potential clinical relevance as in the case of congenital hypogonadotropic hypogonadism.

The question has previously been posed, but the approach used in this manuscript is newly applied and can add to the existing knowledge.

The study is properly designed and performed with the highest technical standards; The data support the authors conclusions.

The conclusions are of interest. Apart from the basic knowledge they expand, they provide potential projections of clinical importance.

The authors address an important question with the most appropriate approach.

Answer. Thank you so much for your positive comments on our study and the nice words you wrote.